# Insights into White Matter Defect in Huntington’s Disease

**DOI:** 10.3390/cells11213381

**Published:** 2022-10-26

**Authors:** Yize Sun, Huichun Tong, Tianqi Yang, Li Liu, Xiao-Jiang Li, Shihua Li

**Affiliations:** Guangdong Key Laboratory of Non-Human Primate Research, Guangdong-Hongkong-Macau Institute of CNS Regeneration, Jinan University, Guangzhou 510632, China

**Keywords:** Huntington’s disease (HD), white matter, oligodendrocyte, myelin, HD animal model

## Abstract

Huntington’s disease (HD) is an autosomal-dominant inherited progressive neurodegenerative disorder. It is caused by a CAG repeat expansion in the Huntingtin gene that is translated to an expanded polyglutamine (PolyQ) repeat in huntingtin protein. HD is characterized by mood swings, involuntary movement, and cognitive decline in the late disease stage. HD patients often die 15–20 years after disease onset. Currently, there is no cure for HD. Due to the striking neuronal loss in HD, most studies focused on the investigation of the predominantly neuronal degeneration in specific brain regions. However, the pathology of the white matter area in the brains of HD patients was also reported by clinical imaging studies, which showed white matter abnormalities even before the clinical onset of HD. Since oligodendrocytes form myelin sheaths around the axons in the brain, white matter lesions are likely attributed to alterations in myelin and oligodendrocyte-associated changes in HD. In this review, we summarized the evidence for white matter, myelin, and oligodendrocytes alterations that were previously observed in HD patients and animal models. We also discussed potential mechanisms for white matter changes and possible treatment to prevent glial dysfunction in HD.

## 1. Introduction

Huntington’s disease (HD) is an autosomal-dominant neurodegenerative disorder characterized by progressive motor symptoms, neuropsychiatric manifestations, and cognitive impairment. The genetic cause of HD is an expanded CAG trinucleotide repeat (>35) in the *HTT* gene encoding an abnormally long polyglutamine (polyQ) tract in the huntingtin (HTT) protein [1,2]. The disease is fatal approximately 15 to 20 years after clinical onset. Therefore, there is an urgent need to develop early diagnosis and intervention method.

HTT is a large protein of 3144 amino acids that can interact with numerous intracellular interactors, which allows HTT protein to participate in diverse cellular functions, including transcription, intracellular transport, cell metabolism, and homeostasis [3]. Although mutant HTT (mHTT) protein is ubiquitously expressed, striatal medium spiny neurons (MSNs) and cortical projection neurons show selective vulnerability [4,5,6,7]. In addition to the polyQ expansion-mediated loss in function of normal HTT and the acquisition of toxic gain function in neuronal cells [3,8], cell nonautonomous, especially glial effects, in the brains of HD have been appreciated [9,10,11,12]. For example, mHTT binds to transcription factor Sp1 and reduces its association with the promoter of glutamate transporter-1 (GLT-1) in mHTT-expressing astrocytes, leading to decreased expression of GLT-1 and reduced removal of glutamate from the synapse, which causes subsequent excitotoxicity and neuronal death [13]. The white matter of the central nervous system comprises myelinated axons that are wrapped around by myelin produced by oligodendrocytes. In the adult brain, oligodendrocytes are mature, terminally differentiated cells. They form myelin that ensheaths axons and allows for rapid neuronal electrical impulses transmission [14]. As oligodendrocytes play a vital role in myelin synthesis and remyelination processes, changes in their numbers or function could impact myelin formation [15,16,17]. 

Dysfunction of oligodendrocytes and myelin is involved in a variety of neurological diseases. In Alzheimer’s disease (AD), oligodendrocyte dysfunction results from direct Aβ toxicity, which could play an important role in the progress of disease, and myelin breakdown contributes to age-related slowing in cognitive function [18,19]. In amyotrophic lateral sclerosis (ALS), degenerative changes in oligodendrocytes were abundantly present in the cortex of human patients and mouse models [20]. The mechanism was suggested due to the reduced level of monocarboxylate transporter (MCT-1), a key player for the shuttling of lactate in and out of oligodendrocytes [21]. Multiple sclerosis (MS) is a common neurological disease in which oligodendrocytes are reduced to varying degrees in demyelinated lesions [22], but the pathogenesis of MS is quite complicated and still unclear, as previous reports postulated that the occurrence of MS might be related to many factors such as heredity, environment, virus infection, and autoimmune disease. In HD, impaired white matter signal was reported in patients at early stage of disease [23,24], and demyelination and axonal damage were found in HD mouse models. 

In this review, we focus on the white matter defects observed in HD patients and animal models. Furthermore, we discuss the importance of investigating the function of HTT and toxicity of mHTT in oligodendrocytes, which would shed new light on the mechanism of oligodendrocyte-related white matter abnormalities.

## 2. Evidence of White Matter Deficiency in HD Patients

At present, the clinical course of genetically defined HD can be divided into three stages: pre-symptomatic period, prodromal period, and manifest period. The pre-symptomatic period means no clinical motor and cognitive signs or symptoms. In the prodromal period, subtle signs and symptoms are present. In the manifest period, clinical motor and/or cognitive impairment can have obvious impact on life [25]. Before the occurrence of motor onset, the prodromal period lasts approximately 15 years [26]. Since there is no established and specific HD biomarker to predict disease progression, patients and families undoubtedly suffer from the uncertainty of the early disease stages. However, the use of neuroimaging can provide a noninvasive, qualitative method of assessing the progression of the disease at an individual level, which is more sensitive and objective than using clinical symptoms as a diagnostic tool. 

The most commonly used imaging methods include magnetic resonance imaging (MRI) and diffusion tensor imaging (DTI). The corresponding analysis methods are voxel-based morphometry (VBM) and tract-based spatial statistics (TBSS). MRI can be used to quantify gray matter volume, white matter volume, whole brain volume, and cerebrospinal fluid (CSF) volume. Since fractional anisotropy (FA) is commonly used as an important index in DTI, FA value can be used to study defects related to microstructural organization. Diffusion of water molecules within white matter is highly anisotropic due to the strong organization of highly myelinated fibers tracks, providing high FA values.

In studies of clinical HD patients and autopsy findings, there is corroborating evidence that macroscopic changes in white matter do occur in the whole brain of HD patients [27,28,29]. Importantly, white matter atrophy often precedes motor symptoms [30,31]. The progression of white matter atrophy occurs in a similar manner to that of gray matter. Using MRI study, Rosas et al. reported that the white matter volume decline in HD patients is age-dependent, and in the pre-symptomatic patient group, white matter volume decrease was not as significant as that in the late stage HD, though there were signs of white matter decrease [32]. Whether the white matter changes in pre-symptomatic patients are significant is controversial. Ciarmiello et al. used MRI evidence to show that asymptomatic mutation carriers have a significantly smaller ratio of white matter to intraventricular volume than healthy controls [33]. The conflicting reports of the white matter changes in HD are plenteous. Some arguments are about whether white matter changes occur randomly, globally, or regionally. Rosas et al. found that accompanied by aggravation of HD, FA decrease first appears in posterior limb of the internal capsule, sub-gyral frontal white matter; with white matter in the sensorimotor cortex, thalamus, and primarily pulvinar. The abnormal white matter image was then found in various brain regions [32]. This study suggests that the white matter changes in the brains of HD patients do not appear to be random, but show a sequential shift. In 2008, Tabrizi et al. also provided evidence by using VBM that white matter in the frontal regions is damaged preferentially during the early stages of the disease [30]. They further reported that white matter loss progresses from the peri-striatal, corpus callosum, and posterior white matter tracts to widespread loss throughout the brain [34]. As for why some regions of white matter appear to be more vulnerable at an earlier stage, McCollgan et al. proposed that the topological length of white matter connections determines their vulnerability in early HD, based on the finding that mHTT causes metabolic disorders through mitochondrial dysfunction [35]. They also found that the abnormal transcription of synaptic genes and metabolic disturbance may drive white matter loss in premanifest HD [36]. 

It would be interesting to know whether myelin changes are secondary to neuronal degeneration. Mouse model studies suggest that loss of the normal fine structure of myelin can cause late-onset axonal degeneration [37]. Our research also shows that there are decreased myelin protein levels in the HD mouse brain and HD patient brain tissues [38]. Imaging study shows that the white matter tends to become smaller and that myelin sheath decomposes in the brains of pre-symptomatic HD [23,24]. By microscopic counting of cell types in the HD brain caudate tissue sections, however, Myers et al. reported that the absolute numbers of oligodendrocytes increased in HD grades 0, 1, and 2 to nearly double of that found in control cases. This report shows that oligodendrocyte density increase does not correlate with neuronal number decrease, suggesting that the increase in oligodendrocyte density is not the result of proliferation in response to neuronal degeneration [39]. In respective to the previous study, Bartzokis et al. proposed a hypothesis that HD pathogenesis begins with a deleterious effect of the mHTTon myelination. Myelin breakdown results in the failure of myelinated neuronal axon transmission, which leads to the excitotoxicity of underlying neurons. In the early stages of the disease, oligodendrocytes may repair and regenerate myelin by increasing their numbers [40].

## 3. Evidence of White Matter Defects in Mouse Models

Since HD is caused by a single gene mutation, HD animal models harboring this genetic mutation would be important for in vivo study [41]. Based on how the HD mouse models were engineered, genetic mouse models were divided into three broad categories: N-terminal HD transgenic models, full-length transgenic models, and knock-in models. These animal models used different promoters to express different lengths of human or murine HTT protein with different polyQ repeat lengths, depending on the needs of the study [42,43]. Changes in white matter and/or myelin were observed in the mouse models established by all three strategies (Table 1).

### 3.1. Full-Length Transgenic Models

Using the yeast artificial chromosome (YAC) or bacterial artificial chromosome (BAC)-mediated transgenic approach, researchers generated full-length transgenic HD mouse models that express full-length mutant human HTT [46,58]. Carroll et al. reported that the corpus callosum volume decreases progressively in the YAC128 mice [44]. MR-DTI measurement shows that YAC128 and BACHD transgenic mice have abnormal white matter structure before behavioral abnormalities and neuronal atrophy appear. Electron microscopy (EM) study reveals thinner myelin sheaths in the corpus callosum of young YAC128 and BACHD mice. Transcript levels of some myelin-associated proteins also show significantly lower levels in the cortex and striatum of YAC128 and BACHD mice [45]. Bardile et al. further showed that BACHD mice exhibit thinner myelin and decreased myelin compaction as early as 1 month of age, suggesting that myelin abnormalities in HD are an early pathological event [47]. Consistently, the specific inactivation of mHTT in OPCs could improve anxiety-like behavior and depressive-like behavior [47].

### 3.2. N-Terminal Transgenic HD Models

N-terminal HD transgenic models develop symptoms more rapidly than full-length and knock in mice. R6/2 mice are the most extensively studied HD mouse model and express a human HTT exon1 with 135 CAG repeats, driven by the human *HTT* promoter. R6/2 mice have the most severe phenotype, though the expression of the mHTT transgene was reported to be at 75% of the endogenous HTT level [49]. Zacharoff et al. quantitatively monitored metabolites in the cortex and striatum of R6/2 mice from one to four months of age by magnetic resonance spectroscopy (MRS). They found that in both brain regions, glycerophosphorylcholine and phosphocholine (GPC and PC) are increased, which suggests there is myelin breakdown [50]. Benraiss et al. analyzed the EM images of the corpus callosum of R6/2 mice. At 6 weeks, the g-ratio of myelinated axons in R6/2 mice do not significantly differ from that of WT mice, while by 12 weeks, the average g-ratio of myelinated callosal axons in R6/2 mice is significantly higher than that of WT controls, suggesting that axons in R6/2 mice have fewer and/or thinner myelin wraps per axon than WT mice. Proteomic analysis shows diminished myelin protein expression, including myelin basic protein (Mbp), myelin-associated glycoprotein (Mag), myelin oligodendrocyte glyco-protein (Mog), myelin oligodendrocytic basic protein (Mobp), proteolipid protein 1 (Plp1), G protein coupled receptor 37 (Gpr37), aspartoacylase (Aspa), and 2,3-cyclic nucleotide-3-phosphodiesterase (Cnp), in 12 week old R6/2 callosal white matter [51]. Rodolfo et al. used UHFD-dMRI and imaging diffusion techniques to examine R6/1 mice (a mouse model made at the same time as R6/2 with the same construct for transgenic process, but with about 115 CAG repeats and the onset of time is much later than R6/2 mice), and they also found reduced connectivity in the mouse brain [59]. Huang et al. generated the *PLP*-HD transgenic mice, which express N-terminal 208 amino acids of HTT with 150Q selectively in oligodendrocytes and found that these mice had an early onset of phenotypes with severe myelin degeneration, suggesting an autonomous mHTT effect in the oligodendrocytes [53].

### 3.3. Knock-in HD Mouse Models

Knock-in (KI) mice are generated by knocking in a human *HTT* exon 1 with an expanded CAG tract into the endogenous mouse *HTT* gene locus (such as the CAG140KI) or only knocking in an expanded CAG repeats [such as the Hdh (CAG) 150)] [54,60,61]. However, HD KI mice express full-length mutant HTT protein at the endogenous level, but have longer life span, milder and much later-onset of phenotypes. Study of white matter in the HD KI mice did not yield consistent results, perhaps because of the mild phenotypes that cannot be readily detected. Perot et al. reported that there was no variation of corpus callosum volume in the CAG140KI mice by using DTI assay, which showed that FA values in CAG140KI mice are only reduced in the first 8 months compared to WT. This difference becomes rather insignificant over time [55]. Others also reported the absence of white matter changes in the Hdh (CAG) 150 mouse brain using MRI study [56]. However, Jin et al. reported early white matter abnormalities in the HdhQ250 mouse brain. [57] Such discrepancies could be due to different numbers of polyglutamine carried by the knock-in mouse model and different disease stages. 

## 4. Evidence of White Matter Defects in Large Animal Models of HD

Compared with rodents, non-human primates (NHPs) are more similar to humans in terms of physiology, metabolism, and brain structure [62]. Unlike the rodent’s brain, monkey’s cerebral structure and composition are more complex. In the monkey brain, the cortex is highly folded into diverse gyri and sulci; the striatum is divided into separate caudate and putamen, and there are thicker and more abundant projection fibers, association fibers, and commissural fibers. Monkeys have more advanced learning abilities that can be used to assess more sophisticated complex movements and cognitive activity [63,64].

It is difficult to obtain NHP models carrying spontaneous mutations that mimic human genetic diseases. Therefore, genetic means such as transgenic expression by using lentiviral vector-mediated transgene expression and CRISPR-Cas9 targeting technology could help facilitate creation of NHP genetics disease models. The CRISPR-cas9 technology has a great potential in precisely modifying the endogenous *HTT* gene of large animals. For instance, a HD knock-in pig model was established and replicated selective neurodegeneration seen in HD patients [65]. However, it is still difficult to establish HD knock-in models of NHPs due to their long sexual maturation time, low reproductive rate, high cost and inefficiency for precise gene editing in monkey models [42]. 

Although no polyQ tract knock-in monkey models are available, transgenic monkey models are valuable for investigation. A HD rhesus monkey was generated by using lentiviral-mediated transgenic *mHTT*, which carried exon 1–10 of the human *HTT* gene with about 70 CAG repeats under the control of human *HTT* promoter [66]. Meng et al. used DTI and TBSS to examine white matter integrity in transgenic HD monkeys every 6 months until 48 months old [66]. Their results show that the FA values of HD monkey (DTI measurement index) reach peak earlier than control monkeys and that peak FA values of HD monkey are generally smaller than control monkey across the whole brain. The abnormal white matter patterns are observed not only in the fibers connecting the cortical areas to the caudate and putamen, but also in multiple fiber tracts and a few cortical areas [66]. White matter tract changes may reflect an earlier demyelination process, while gray matter area may be affected by earlier astrocytosis or microglia remodeling during white matter development [66]. The generation of the HD transgenic monkey has certainly shown great advantages in the longitudinal observation of white matter lesions in HD. Weiss et al. generated monkey models by injecting viruses expressing N-terminal mutant HTT (N171-85Q) into the adult monkey caudate and putamen, resulting in mHTT expression throughout the entire macaque cortico-striatal circuit [67]. They used DTI to image white matter fiber tracts a few months later after surgery. Imaging shows that the N171-85Q monkey exhibits more significant changes in the FA values in the dorsal and ventral prefrontal white matter tracts, as well as the anterior/dorsal corona radiata, internal capsule, external capsule, and corpus callosum [67]. Viral mHTT-expression-generated HD monkeys can be produced in a relatively short time and in sufficient numbers to test treatments, and to find novel biomarkers, though this model is not genetically transmissible.

In addition to NHPs models, transgenic sheep and pig models were also created. OVT73 is a transgenic HD sheep model expressing full-length human HTT with 73Q [68]. Taghian et al. report that the FA value of white matter decreases in the HD sheeps, while the metabolite index (GPC + PC) of demyelination increases in the aging stage of OVT73 sheep [69]. Pigs are also suitable for studying degenerative diseases, because they have rich white matter similar to human brains [70]. Baxa et al. generated (TgHD) minipigs encoding human huntingtin (HTT)1-548 under the control of human *HTT* promoter [71]. Ardan et al. found decreased myelination of nerve fibers in the internal capsule and in the subcortical white matter, while the TgHD minipigs’ motor and cognitive abilities decreased [72]. In the study of hoofed HD animal model, the process of behavioral changes shows gender specificity [69,72].

## 5. Mechanism of Myelin Defects in HD

Myelin abnormalities have been linked to two putative pathways in HD (Figure 1). One is the effect of mHTT on the normal function of MYRF (myelin regulatory factor), and the other is the down-regulation of PGC1α (peroxisome-proliferator-activated receptor gamma coactivator 1 α). 

### 5.1. MYRF in HD 

Differentiation of the progenitor of oligodendrocytes (OPC) is controlled by a core regulatory network consisting mainly of OLIG2, SOX10, NKX2.2, ZFP24, and MYRF transcription factors [73]. Among them, MYRF is in the downstream of the regulatory network, and regulates the expression of genes encoding important protein components of myelin including *Mbp*, *Plp1*, *Mag*, and *Mog*, etc. [74].

MYRF is a membrane-bound transcription factor localized in the ER(Endoplasmic reticulum) that can release an N-terminal fragment (nMYRF) containing a DNA-binding domain after self-cleavage. nMYRF is able to enter the nucleus to perform transcriptional regulatory functions by recognizing specific sequences in the promoter regions of genes such as *MBP, MAG*, etc. [75]. Our earlier study shows that mHTT binds to nMYRF and affects its transcription activity by altering its association with DNAs, resulting in reduced myelin gene expression and oligodendrocyte dysfunction [53]. MYRF expression and phosphorylation levels are associated with oligodendrocyte pathology in HD. Using RNA-Seq, Benraiss et al. analyzed striatal OPCs that were acutely isolated from HD transgenic mice. Their study shows that several myelinogenic genes including *Myrf* in both R6/2 and zQ175 mice are downregulated compared with their littermate controls [51]. The downregulation of *Myrf* is also found in human HD hESC-derived GPCs (glial progenitoe cells) [76]. Our recent study shows that the reduced MYRF phosphorylation inhibits MYRF’s binding to mutant huntingtin and increases the expression of myelin-associated genes [38], suggesting that MYRF phosphorylation inhibition could be a druggable target for treating white matter degeneration in HD. 

### 5.2. PGC1α in HD 

PGC1α is a transcriptional coactivator. Cui et al. performed studies in PGC1α-KO mice to investigate the relationship between abnormal energy metabolism and PCG1α in HD, which showed that PGC1α regulates the level of MBP and rate-limiting enzymes mRNA for cholesterol synthesis [77]. Xiang et al. used primary oligodendrocytes culture and R6/2 mice to show that mHTT reduced PGC1a and affected its transcriptional activity on MBP and cholesterol synthesis, which resulted in demyelination in HD [48].

## 6. Discussion

As selective neuronal loss and degeneration are the most striking pathological features of HD, a large body of work focused on probing into neuronal abnormalities such as synaptic dysfunction, mitochondrial toxicity, and axonal transport impairment caused by mHTT in neurons [26]. However, mHTT is present in all cell types in the brain, and its expression in glial cells can also contribute to HD neuropathology [12]. Although *mHTT* carriers can be identified by genetic testing, there is still no good biomarker that can predict the progression of premanifest HD. However, white matter changes can occur prior to the clinical onset and are increasingly recognized as a reference marker for the early diagnosis of HD. In addition, oligodendrocytic and myelin dysfunction are shown to link with mood disorders and depression [78,79,80]. Since the depressive symptoms in HD patients appear before the manifestation of motor symptoms [81], using brain imaging to monitor white matter changes would be a good clinical indicator for HD progression.

So far, there are few therapies aiming to improve white matter pathology in HD models. Laquinimod (LAQ) is an immunomodulatory agent for alleviating demyelination in MS [82]. LAQ may restore the expression of myelin genes by reducing the phosphorylation of MYRF [38]. It is shown to rescue striatal, cortical, and white matter pathology and improve behavioral phenotypes in the YAC128 mouse and the *PLP*-150Q mice [38,83]. In addition, gene therapy approaches have considerable potential in treating certain neurodegenerative diseases [84]. There have been many strategies using gene therapy for HD in the past, including ASO (antisense oligo), siRNA, shRNA, and miRNA [85,86,87,88]. However, little is reported about their rescue effects on white matter.

When comparing white matter pathologies in HD patients and animal models, the tremendous benefits of modeling using NHPs are revealed. Rodent and primate brains are noticeably different in many aspects. From a cell lineage and development perspective, primates develop an enlarged cortical germinal zone called the outer subventricular zone (OSVZ) that is absent in rodents but is an additional source of OPCs [89]. From the anatomical standpoint, in rodent models, the only areas of white matter where changes can be measured are the corpus callosum, while human white matter is ~3000 times larger than the rodent counterpart. The enlargement of white matter in the frontal lobe represents the evolutionary advance of primates [90,91]. The reported preferential changes in frontal white matter in both HD patient and transgenic monkey brains could be linked with the patient’s early mental symptoms such as poor attention, irritability, impulsivity, and poor mood regulation [92].

Brain cholesterol accounts for a considerable proportion of total body cholesterol, and oligodendrocyte myelin consists of most of the brain cholesterol content. Cholesterol biosynthesis is a sophisticated and delicate process governed by many enzymes. Little is known about the role of cholesterol dysregulation in HD. Kacher et al. reviewed a number of previous studies in the field of altered cholesterol homeostasis in HD [93]. Interestingly, increased cholesterol levels were found in the striatum of HD patients autopsied [94], which could explain the increased number of oligodendrocytes in the HD postmortem striatum [39].

How normal HTT protein functions in oligodendrocytes is not very clear, though there has been some evidence for the contribution of mHTT expression to oligodendrocyte pathology. Whether white matter abnormalities are a cause or a consequence of HD axonal lesions is not clear, but alterations in white matter and oligodendrocytes should not be separated from neuronal pathology, as there is growing evidence that the axonal myelination is dependent on the interaction between axons and oligodendrocytes and is closely linked to intracellular signaling pathways [95]. HTT has been shown to interact with other proteins to function as a large scaffolding protein involved in axonal transport, and mHTT-mediated alterations in synaptic transport in nerve cells may also alter the myelin sheath development and function. To understand the HD neuropathogenesis, further studied to delineate HTT function and toxicity of mHTT in the oligodendrocytes and other types of glial cells need to be conducted.

## Figures and Tables

**Figure 1 cells-11-03381-f001:**
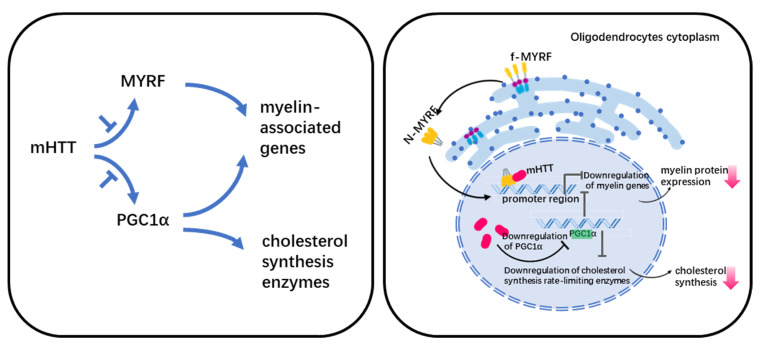
Legend. Potential mechanism of myelin defect in HD.

**Table 1 cells-11-03381-t001:** HD mouse models used for myelin defect studies.

Animal Model	Construct	Alterations in White Matter Structure	g-Ratio	Axonal Microstructure	Expression of Myelin Proteins	Reference
YAC128	FL human *HTT* with 128 mixed CAA/CAG	CC volume decrease from 3 months FA decrease in AC, CC, EC, et al.	Increase from 1.5 months		RNA transcripts of myelin-related genes decrease	[44,45]
BACHD	FL human *HTT* with 97 mixed CAA/CAG	FA decrease in AC, CC, EC et al.	Increase from 1 month	Less compact myelin	RNA transcripts of myelin-related genes decrease. Reduced expression of MBP In the striatum at 2 weeks	[45,46,47,48]
R6/2	Express exon1 mutant hHTT under *HTT* promoter	DTI shows FA values decreased significantly in CC at 12 weeks old	Increase at 12 months	A reduction in mean callosal axon caliber	Reduced expression of MBP in the CC and striatum from P14	[48,49,50,51,52]
*PLP*150Q	Express N-terminal mHTT under *PLP* promoter		Increase from 3 months of age	Decreased myelination of striatal axons from 3 months	Reduced expression of CNP, MBP, MOBP and MOG in the brain stem	[53]
CAG140KI	Knock in human *mHTT* exon1 into mouse HTT locus	A delay in FA increase in first 8 months				[54,55]
Hdh150Q	Knock in CAG tract in mouse *HTT* locus	No significant difference in DTI index in cc white matter		No significant difference in CC axon density		[56]
Hdh250Q	Knock in CAG tract in mouse *HTT* locus		Increase at 12 months	Significantly fewer myelinated axons in CC	Reduced expression of MBP and MOP in the striatum at P14	[57]

Note: g-ratios of myelinated axons, a measure of myelin sheath thickness calculated as the ratio of axon diameter (axon caliber) to myelinated fiber diameter. Abbreviations: AC: anterior commissure; CC: corpus callosum; CNP: 2′,3′-cyclic nucleotide 3′-phosphodiesterase; DTI: diffusion tensor imaging; EC: external capsule; FA: fractional anisotropy; FL: full-length; MBP: myelin basic protein; MOBP: myelin-associated oligodendrocytic basic protein; MOG: myelin oligodendrocyte glyco-protein.

## Data Availability

Not applicable.

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
