# Peer review of "Insights into White Matter Defect in Huntington’s Disease"

_cells, 2022, doi:10.3390/cells11213381_

Round 1

Reviewer 1 Report

There is accumulating evidence that demonstrate that white matter and in particular oligodendrocytes are affected in Huntington's disease contributing to the clinical manifestations. One could hypothesize that white matter defects and non-motor symptoms constitute early biomarkers of disease trajectory of crucial interest to consider in clinical trials targeting mHTT. Sun et al. give an overview of our current knowledge in this important area of research. This review comes from an excellent research group that has published significant papers on glia and oligodendrocytes dysfunction in HD. Overall, the paper is comprehensive including the important (published) studies in this research area and adequately discuss the current state. This will be a very useful review for researchers working on white matter changes in neurodegenerative diseases.

There are a few language errors that should be corrected:

Page 1 line 14 ,clinical image’ please change to ,clinical imaging’

Page 1 line 14 ,white matter abnormal‘ please change to ,white matter abnormalities‘

Page 2 line 61: “(23,24 inserted)“,  “inserted “ should be removed

Page 2 line 97 “The conflict reports”, please change to “conflicting”

Page 5 line 222 “They used DTI to imagine white matter fiber tracts“, please change to “image”

Resolution of Fig. 1 may need to be improved (in the pdf file it seems to be a bit blurry and hard to read)

Page 6 line 239/240 “Among them, MYRF is in the downstream of the regulatory network whose binding sites were present…“- in this sentence it could be understood that “whose” is referring to “the network”. Maybe should be changed to „MYRF is in the downstream of the regulatory network and its binding sites are present…”

Page 6 line 243 “after self-extrude”, this should be reworded, maybe the authors meant ‘self-extrusion’?

Page 6 line 264 “which resulted demyelination in HD” – please change to “which resulted in demyelination in HD”

Page 7 line 273 “prior to the clinic onset“ – please change to “prior to the clinical onset”

Page 7 line 277 “image to monitor white matter changes” – please change to “imaging to monitor white matter changes”

Page7 line 282 “It has been showed to rescues“ – please change to “It has been shown to rescue”

Page 7 line 284 “gene therapy approach has considerable” –  please change to “gene therapy approaches have considerable”

Page 7 line 292 “primate develop an enlarged” – please change to “primates develop an enlarged”

Page 7 line 306 “HD disease (86).“ – please remove “disease”

Page 7 line 311 “Whether white matter abnormal changes is a cause” – please change to “Whether white matter abnormalities are a cause”

Author Response

We thank the reviewer’s positive comments and appreciate the detailed corrections for language errors. We have corrected all the errors accordingly. We also generated a new figure with better resolution.

Reviewer 2 Report

In this review, the authors show the evidence for white matter, myelin, and oligodendrocytes alterations that were observed in Huntington Disease patients and animal models. In the paper, the authors list different animal models but not all. They show the defect in the white matter of transgenic mice but for example, they do not cite the data obtained in a recent paper on R6/1 mouse model of HD (Gatto RG, Weissmann C. Preliminary examination of early neuroconnectivity features in the R6/1 mouse model of Huntington's disease by ultra-high field diffusion MRI. Neural Regen Res. 2022 May;17(5):983-986). They didn’t show data about the rat models of HD or about other animal models like mini pig (Vidinská D, Vochozková P, Šmatlíková P, Ardan T, Klíma J, Juhás Š, Juhásová J, Bohuslavová B, Baxa M, Valeková I, Motlík J, Ellederová Z. Gradual Phenotype Development in Huntington Disease Transgenic Minipig Model at 24 Months of Age. Neurodegener Dis. 2018;18(2-3):107-119) or sheep (Taghian T, Gallagher J, Batcho E, Pullan C, Kuchel T, Denney T, Perumal R, Moore S, Muirhead R, Herde P, Johns D, Christou C, Taylor A, Passler T, Pulaparthi S, Hall E, Chandra S, O'Neill CA, Gray-Edwards H. Brain Alterations in Aged OVT73 Sheep Model of Huntington's Disease: An MRI Based Approach. J Huntingtons Dis. 2022 Sep 29), only as example.

I think a review should arise from a deeper revision of the topic. Therefore, I suggest expanding the analysis of the literature in this field.

I also suggest a check of typos errors in the manuscript; view raws 61, 230, 297, 306.

Author Response

We thank the reviewer’s comments and suggestions, we have now added all the suggested literature in the revision and discussed them on page 5 section 2. We have carefully corrected the English spelling and grams errors.

Reviewer 3 Report

In this paper, the authors review the currently available data on the pathology of the white matter in the brains of HD patients. Specifically, they have summarized the evidences for white matter, myelin, and oligodendrocytes alterations that have been observed in HD patients and animal models. Also, they discuss potential mechanisms for HD white matter changes and potential treatment to prevent glial dysfunction in HD.

 Comment:

 The paper is well-structured and covers many relevant areas on the topic. A figure is also provided.

It would help the addition of a synoptic table, for consultation to the reader.

A grammar- and spell-check is advised.

Author Response

(The authors gave the same response as above.)

Reviewer 4 Report

In this manuscript, Sun et al. give insights into white matter defects observed in Huntington Disease. They focuse on the white matter defects observed in HD patients and rodent and non-human primate animal models. They also discuss the importance of investigating the function of huntingtin protein and toxicity of mutant huntingtin in oligodendrocytes, which would shed new light on the mechanism of oligodendrocyte-related white matter abnormalities.

Finding an early sign or a biomarker is important in terms of presence of a long pre-symptomatic period which reduces the change for an efficient treatment. In this context, this review points out the significant potential of early white matter changes in HD. Besides, they give future directions about further analyzing the oligodendrocyte changes causing white matter abnormalities related to mHTT protein funciton to better understand HD neuropathophysiology.

However, there is a point to be reconsidered:

-       The title ‘Evidence of glial defects in mouse models’ does not seem appropriate because the text mentions white matter rather than glia cells.

Author Response

We thank the reviewer for a positive review of our manuscript and good suggestions. To be more precisely adherent to our review topic, we have changed the title “Evidence of glial defects in mouse models” to “Evidence of white matter defects in HD mouse models“